# Diagnostics to support the eradication of yaws—Development of two target product profiles

**Noah Fongwen[1], Becca L. Handley[1], Diana L. Martin[2], Camila Beiras[3], Louise Dyson[4], Michael Frimpong[5], Oriol Mitja[3], Kingsley Asiedu[6], Michael Marks** [1,7,8]*

**1** Department of Clinical Research, Faculty of Infectious and Tropical Diseases, London School of Hygiene and Tropical Medicine, Keppel Street, London, United Kingdom, **2** Division of Parasitic Diseases and Malaria, Centers for Disease Control and Prevention, Atlanta United States of America, **3** Fight Aids and Infectious Diseases Foundation, Badalona, Spain, **4** The Zeeman Institute for Systems Biology & Infectious Disease Epidemiology Research, School of Life Sciences and Mathematics Institute, University of Warwick, Coventry, United Kingdom, **5** Kumasi Centre for Collaborative Research, Kwame Nkrumah University of Science & Technology, Kumasi, Ghana, **6** Department for the Control of Neglected Tropical Diseases, World Health Organization, Geneva, Switzerland, **7** Hospital for Tropical Diseases, University College London Hospital, London, United Kingdom, **8** Division of Infection and Immunity, University College London, London, United Kingdom

* michael.marks@lshtm.ac.uk

**Data Availability Statement:** All relevant data are within the manuscript.

**Funding:** The authors received no specific funding for this work.

## Abstract

### Background

Yaws is targeted for eradication by 2030, using a strategy based on mass drug administration (MDA) with azithromycin. New diagnostics are needed to aid eradication. Serology is currently the mainstay for yaws diagnosis; however, inaccuracies associated with current serological tests makes it difficult to fully assess the need for and impact of eradication campaigns using these tools. Under the recommendation of the WHO Diagnostic Technical Advisory Group (DTAG) for Neglected Tropical Diseases(NTDs), a working group was assembled and tasked with agreeing on priority use cases for developing target product profiles (TPPs) for new diagnostics tools.

### Methodology and principal findings

The working group convened three times and established two use cases: identifying a single case of yaws and detecting azithromycin resistance. One subgroup assessed the current diagnostic landscape for yaws and a second subgroup determined the test requirements for both use cases. Draft TPPs were sent out for input from stakeholders and experts. Both TPPs considered the following parameters: product use, design, performance, configuration, cost, access and equity. To identify a single case of yaws, the test should be able to detect an analyte which confirms an active infection with at least 95% sensitivity and 99.9% specificity. The high specificity was deemed important to avoid a high false positive rate which could result in unnecessary continuation or initiation of MDA campaigns. If used in settings where the number of suspected cases is low, further testing could be considered to compensate for imperfect sensitivity and to improve specificity. The test to detect azithromycin resistance should be able to detect known genetic resistance mutations with a minimum

**Competing interests:** The authors have declared that no competing interests exist.

sensitivity and specificity of 95%, with the caveat that all patients with suspected treatment failure should be treated as having resistant yaws and offered alternative treatment.

## Conclusions

The TPPs developed will provide test developers with guidance to ensure that novel diagnostic tests meet identified public health needs.

## Author summary

Accurate diagnostic tests are needed to aid yaws eradication efforts. Diagnostic tests are important for determining where yaws is present and for monitoring eradication efforts. Whilst there are tests available, they have limitations and will not all be suitable in all settings, especially as disease prevalence reduces in the move towards eradication. Therefore, new diagnostics solutions are needed. To aid with this, we determined the programmatic areas of greatest need (use cases) and then developed a shortlist of product requirements (target product profiles, or TPPs) for each scenario. These TPPs can then be used by product developers to ensure that novel diagnostic tools in development are fit for purpose. There were two programmatic use cases for which yaws TPPs were developed. The first TPP focused on diagnostics to detect a single case of yaws in a community, thus highlighting the need for, or continuation of mass drug administration efforts. The second TPP lays out the requirement for a test that can detect resistance to azithromycin, the antibiotic used for the eradication campaigns. This will be key to rapidly detect emergent antibiotic resistant bacteria and prevent it from being passed on.

## Introduction

Yaws is a bacterial infection caused by infection with *Treponema pallidum* subsp. *Pertenue.*. More than 80 countries and territories are known to have previously reported or currently report cases of yaws. The disease is now predominantly reported in West and Central Africa and the Pacific but has previously been endemic throughout the tropics [1,2]. Most cases of yaws are found in children. Transmission is believed to occur primarily through skin-to-skin contact with infectious lesions of yaws. *T.pallidum* subsp *pertenue* is closely related to *T.pallidum* subsp. *pallidum*, the causative agent of syphilis, but unlike the latter is not transmitted through sexual contact or from mother to the developing foetus during pregnancy.

The clinical course of yaws starts with primary yaws, characterised by a single infectious lesion. If the individual is untreated, the initial lesion will heal but the infection will continue. This asymptomatic state is called latent yaws [3]. The individual may then develop secondary yaws, during which time new infectious lesions occur. The patient may alternate between asymptomatic latent yaws and further episodes of secondary yaws for several years. If patients remain untreated for a long time (>5 years), they may develop tertiary yaws, characterized by lesions that are destructive and disfiguring but not infectious.

Yaws has been targeted for eradication since the mid-20<sup>th</sup> century. Previous efforts focused on the use of mass or targeted treatment with penicillin. Although these efforts reduced the prevalence of yaws substantially they were ultimately not successful at achieving eradication [1]. In 2012, the World Health Organization (WHO) launched a new strategy—'the Morges strategy'—based on mass treatment with azithromycin [4]. Central to this strategy is the identification of all endemic communities, followed by community mass treatment of the whole

population. This is known as total community treatment or TCT. Implementation of this strategy has been piloted on a small scale in several settings.

Once the number of yaws cases has fallen, the strategy switches to total targeted treatment (TTT) which involves ongoing case finding to identify and treat remaining cases and their close contacts. Once no further cases of yaws are detected, the programme switches to surveillance, including serological surveillance of children aged <5 years [4].

As the aim of the programme is eradication, detection of single cases is important. This occurs in 2 scenarios: i) deciding whether to initiate a programme–in this setting finding a single yaws case is sufficient to declare a community endemic and needing treatment; and ii) surveillance after mass treatment–in this setting it is important to ensure that there is not even a single suspected case that is truly yaws so that interventions can safely stop.

A major concern has been the identification of a small number of cases of yaws which have developed resistance to azithromycin [5]. In Papua New Guinea, long term follow up studies have reported Azithromycin resistant yaws cases 24 months post MDA [6,7]. Two point mutations (A2059G and A2058G in the 23S ribosomal genes) are known to confer resistance to azithromycin in *T.pallidum* subsp *pallidum*, and both have now has been detected in *T.pallidum* subsp *pertenue*. Detection of phenotypic resistance is complicated by the inability to routinely culture *T. pallidum*, and therefore the focus of resistance detection is on a) identification of cases of clinical treatment failure, and b) identification of known mutations associated with resistance to azithromycin.

The availability of point-of-care tests and PCR testing facilities for yaws remains a critical issue for all countries to reach 2030 yaws eradication target as outlined in the 2021–2030 Neglected Tropical Diseases (NTD) roadmap [8]. WHO's Department of Control of Neglected Tropical Diseases manages a diverse portfolio of diseases, each with its own unique epidemiological and diagnostic challenges. It was decided by the Strategic and Technical Advisory Group (STAG), the principal advisory group to the Director-General of WHO on the control, elimination and eradication of NTDs, that a single WHO working group would help ensure that a unified approach could be used to identify and prioritize diagnostic needs, and to inform WHO strategies and guidance on the subject [9].

The first meeting of the Diagnostic Technical Advisory Group (DTAG), an advisory group to the Department of Control of Neglected Tropical Diseases, was held in Geneva, Switzerland, on 30th and 31st October 2019. DTAG members discussed priorities for the year ahead as well as how to manage the complexity of supporting the diagnostics agenda across the entirety of WHO's NTD portfolio. Recommendations were made, based on the understanding that they would be reviewed at the next meeting, as it had been made clear that all NTDs had diagnostic needs which would have to be addressed in due course. One of the recommendations was that target product profiles (TPPs) for diagnostics were needed to support emerging yaws control and eradication programmes.

In this manuscript, we report the process used in developing the TPPs for yaws, the TPP specifications and the assumptions made. The purpose of these TPPs is to support the yaws eradication strategy by describing the preferred and the minimally acceptable profiles for diagnostics for 'detection of a case of yaws' and 'detection of an azithromycin-resistant case of yaws'. In doing so the TPPs provide guidance to test developers as to what test characteristics are likely to be required for a test to be adopted as part of the yaws eradication strategy. For detection of a case of yaws, the tool or tools must be able to detect a single case of active yaws infection. For detection of a case of azithromycin-resistant yaws, the tool or tools must be able to detect mutations known to be associated with azithromycin resistance in yaws.

## Methods

Upon recommendation of the DTAG, WHO formed a group of skin NTD experts, clinicians and other stakeholders. Different subgroups for skin NTDs were formed including the yaws subgroup. The seven member yaws subgroup met from January 2021 to April 2021 to agree on priority use cases for the TPPs and execute the process for the developments of the TPPs. The two priority use cases for yaws were: identifying a single case of yaws (case detection) and detecting azithromycin-resistant yaws. Two expert subgroups were formed, one to determine the attributes required for each use case (use case characteristics) and another to conduct landscaping of the available diagnostic assays. TPPs were intended to help guide test developers so that any newly developed diagnostic assays were capable of addressing prioritized public health needs. Using the WHO core TPP development process (Fig 1), the expert subgroups for yaws convened online a total of three times to discuss and determine attributes required for each use case.

TPPs for each use case considered the following parameters: product use, design, performance, product configuration and cost, and access and equity. Initial Draft 0 requirements in each TPP were selected based on landscape analyses, use case needs analysis and diagnostics performance modeling developed through a consultative process coordinated by the WHO Department of the Control of Neglected Tropical Diseases. For certain elements in each use case, parameters were defined at the outset, and assumptions were made to move forward with sensitivity and specificity calculations. These included recognizing that reaching eradication of yaws requires detection of individual cases, that repeated testing would be possible for suspected cases when case numbers approach zero, and that additional tools such as sequencing may be needed for case confirmation in some circumstances. For the azithromycin resistance TPP a key consideration was the availability of a safe alternative treatment in the form of injectable benzathine-penicillin. The yaws subgroup critically reviewed and modified the zero draft where warranted. The zero draft was sent to the DTAG for review and comments.

After revising based on the comments from the DTAG, the yaws subgroup finalized the TPP details, and draft 0.1 TPPs were posted on the WHO website for public comment in July 2021. Comments received were shared with the experts, and TPPs were revised accordingly to generate version 1.0 TPPs.

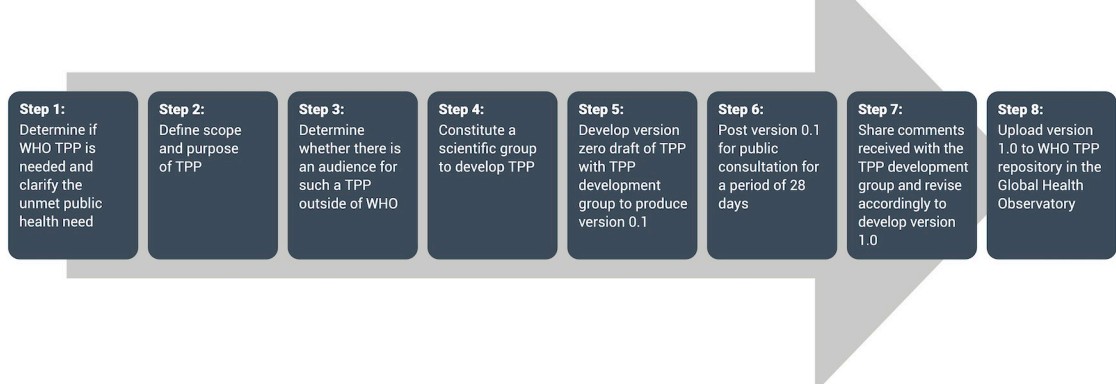

**Fig 1. World Health Organization Target Product Profile Process.**

**Table 1. Select characteristics of needed test for case detection.**

| Feature | Ideal requirement | Minimum requirement |
|---|---|---|
| Intended use | An in vitro point-of-care test that detects *T pallidum pertenue*-specific analyte(s) for the purpose of identifying a single case of yaws and confirming active infection to know if an area is endemic for yaws. | An in vitro laboratory-based test that detects *T pallidum*-specific analyte(s) for the purpose of identifying a single case of yaws and confirming active infection to know if an area is endemic for yaws. |
| Target analyte | Biomarker(s) specific for active infection with *T pallidum pertenue*. | Biomarker(s) specific for current active or latent infection from *T pallidum*. |
| Diagnostic/ clinical sensitivity[a] | >95% | >95% |
| Diagnostic/ clinical specificity[b] | >99.9% | >99.9% |

[a] In situations where the number of suspected cases being tested is low (less than 5), further testing could be considered such as repeated testing or testing for a second target could be considered to compensate for imperfect sensitivity.

[b] In situations where the number of positives is low(less than 5), further testing could be considered such as testing for a second target or sequencing to improve specificity.

## Results

Version 1.0 TPPs for the two use cases were published by WHO on 7 November 2021 within the WHO Global Observatory on Health R&D. Select TPP features and their associated requirements are presented in Tables 1 and 2.

## Discussion

The yaws eradication strategy (Morges strategy) recommends treatment of an entire community with either azithromycin or injectable penicillin with a minimum treatment coverage of

**Table 2. Select characteristics of needed test for detection of azithromycin-resistant yaws.**

| Feature | Ideal requirement | Minimum requirement |
|---|---|---|
| Intended use | An in vitro point-of-care test that detects *T pallidum pertenue*-specific analyte(s) for the purpose of detecting azithromycin resistance in yaws patients with active infection caused by currently known mutations | An in vitro laboratory-based test that detects *T pallidum*-specific analyte(s) for the purpose of detecting azithromycin resistance in yaws patients with active or latent infection caused by currently known mutations. |
| Target analyte | Biomarker(s) specific for the DNA of azithromycin-resistant *T pallidum pertenue* (azithromycin resistance genes) in those with active yaws infection. | Biomarker(s) specific for the DNA of azithromycin-resistant *T pallidum pertenue* (azithromycin resistance genes) in those with active or latent yaws infection. |
| Diagnostic/ clinical sensitivity[c] | >99% | >95% |
| Diagnostic/ clinical specificity[d] | >99% | >95% |

[c] In the absence of perfect sensitivity, programmes will need to presumptively treat cases that fail first line treatment as cases of resistant yaws.

[d] Recognition that in the absence of perfect specificity some patients will receive benzathine penicillin but this has previously been standard of care and is considered acceptable.

90%, upon identification of a single confirmed case of yaws (4). Progress has been made towards the eradication of yaws in some countries but there are diagnostic challenges that may be a barrier to the success of these eradication efforts. Two challenges faced are: the lack of molecular point of care tests and PCR testing facilities; and the emergence of azithromycin-resistant yaws. The availability of tests that can reliably identify a single case of yaws is critical in deciding when to start and stop mass drug administration (MDA) as is the availability of tools that can rapidly and effectively detect cases of azithromycin resistance post MDA will be useful in avoiding unnecessary treatment and ensuring resistant cases can be treated directly with benzathine penicillin [5]. To date, clinical examination combined with serological tests for *T. pallidum* (i.e., not specific for *T.p. pertunue*) remains the mainstay of yaws diagnosis [10]. Available serological tests can be performed either in a laboratory or in the field through POC lateral flow assays. However current serological tests have a number of limitations which means they are not useful for detecting single cases of yaws which is required for eradication programmes. Molecular diagnostic tests such as PCR are commonly used in research and increasingly within programmes. As yaws programmes move towards eradication and with the emergence of drug resistance, the need for better diagnostics has become more urgent. This puts particular pressure on the development of TPPs to lay out specifications that new tests will need to meet to fill the needs of eradication programmes.

Currently, PCR and sequencing can be used to detect genes that confer azithromycin resistance. Challenges for both techniques include the high complexity and training needs, high costs and equipment requirements, and the lack of performance data in those with latent yaws. Additionally, PCR can only detect mutations already known to be associated with azithromycin resistance and only sequencing could detect de-novo mutations in previously undescribed genomic targets. Ideally, the TPP specifies the need for point of care tests that fulfill the WHO assured criteria. These tests should have a very high sensitivity and specificity (close to 100%) to detect known azithromycin resistance mutations in yaws patients with active infection. To scale up case finding and accelerate effort towards eradication, these tests should also have real-time connectivity, use specimens that are easy to collect, and be affordable, sensitive, specific, user-friendly, rapid, equipment-free and deliverable to those who need it. Developing such tests will be challenging but since no test is perfect, programmes need to be prepared to put in place strategies to overcome the sensitivity and specificity shortcomings. In the absence of perfect sensitivity, programmes will need to presumptively treat cases that fail first line treatment. In the absence of perfect specificity, some patients will receive benzathine penicillin when not necessary, which confers some risk to the patient. However, this has previously been the standard of care [11]. No cases of penicillin resistance have been documented to date in either yaws or syphilis.

The need to identify a single case of yaws in a community and confirm active yaws infection in the era of eradication means the test should ideally have very high sensitivity and specificity to ensure programmes make the best use of the resources available. As in the case of azithromycin resistance, strategies need to be in place to reduce errors that may arise due to imperfect sensitivity and specificity. In situations where the number of suspected cases being tested is low (less than 5), further testing could be considered to compensate for imperfect sensitivity. This could involve repeated testing of the same individual or detection of a second PCR target. Conversely, if the number of positives is low (less than 5), detection of a second PCR target or gene sequencing could otherwise be used to improve specificity.

Yaws infection can be active or latent [3]. A limitation of the current TPPs is that they are focused on tests that can be used to detect an active infection of yaws. Even with molecular tests such as PCR, distinguishing latent from active yaws is a major challenge [12]. Diagnosing latent yaws will be important for identifying (and treating) potential cases that may relapse

and propagate transmission as well as preventing the progression of infection to the later stages that are associated with more morbidity and disability. The success of yaws eradication therefore also depends on the availability of point of care tests that can be used in the field to identify cases of latent yaws. Importantly, it should be noted that the recommended TPPs have not yet been tested in any of the populations described. As a result, this paper is hypothetical based on a number of well-established assumptions. Therefore, the recommendations will need to be validated in real-life situations. Despite these limitations, we believe the development of these two TPPs for yaws is an important step towards ensuring that new tests are developed to assist yaws programmes to achieve eradication. Future efforts will need to be directed towards developing TPPs for latent yaws, and for tests that can be used for certification of elimination and post elimination surveillance.

## Conclusion

Two TPPs for yaws assays have been developed by the working group established by the WHO Diagnostic Technical Advisory Group (DTAG) for Neglected Tropical Diseases(NTDs). The first lays out the specifications for a test to detect a single case of yaws which could be used to determine if a community is endemic and whether to initiate or discontinue MDA. This test should be highly specific to avoid unnecessary interventions, and sensitive enough to not miss a single case, so ensuring all endemic communities are targeted with eradication interventions. The TPP for azithromycin resistance detection also has the requirement of high sensitivity and specificity for known mutations in order to successfully detect the majority of resistant cases. Where specificity is not 100%, presumptive treatment with penicillin of anyone with treatment failure should prevent spread of resistant cases.

As well as these technical characteristics, diagnostic tests developed for use in NTD control or eradication programmes should also be designed to fulfill the WHO reassured criteria. This means tests should have real-time connectivity, use specimens that are easy to collect, and be affordable, sensitive, specific, user-friendly, rapid, equipment-free and deliverable to those who need it. Thus, the development of affordable point-of-care tests with high accuracy would be ideal for the yaws eradication effort.

## Acknowledgments

The authors would like to thank all experts and colleagues who provided useful comments through the public consultation.

## Disclaimer

The findings and conclusions in this report are those of the authors and do not necessarily represent the official position of the Centers for Disease Control and Prevention or the World Health Organization.

## Author Contributions

**Conceptualization:** Kingsley Asiedu, Michael Marks.

**Methodology:** Noah Fongwen, Becca L. Handley, Diana L. Martin, Camila Beiras, Louise Dyson, Michael Frimpong, Oriol Mitja, Michael Marks.

**Writing – original draft:** Noah Fongwen, Becca L. Handley, Louise Dyson, Michael Marks.

**Writing – review & editing:** Diana L. Martin, Camila Beiras, Michael Frimpong, Oriol Mitja, Kingsley Asiedu, Michael Marks.

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
