## [Decision Letter · Decision Letter 0]

2 Aug 2022

Dear Dr. Marks,

Thank you very much for submitting your manuscript "Diagnostics to support the eradication of yaws – Development of Two Target Product Profiles" for consideration at PLOS Neglected Tropical Diseases. As with all papers reviewed by the journal, your manuscript was reviewed by members of the editorial board and by several independent reviewers. The reviewers appreciated the attention to an important topic. Based on the reviews, we are likely to accept this manuscript for publication, providing that you modify the manuscript according to the review recommendations. 

Sincerely,

Justin D Radolf

Guest Editor

Richard Phillips

Section Editor

Reviewer's Responses to Questions

**Key Review Criteria Required for Acceptance?**

**Methods**

-Are the objectives of the study clearly articulated with a clear testable hypothesis stated?

-Is the study design appropriate to address the stated objectives?

-Is the population clearly described and appropriate for the hypothesis being tested?

-Is the sample size sufficient to ensure adequate power to address the hypothesis being tested?

-Were correct statistical analysis used to support conclusions?

-Are there concerns about ethical or regulatory requirements being met?

Reviewer #1: Accept

Reviewer #2: The methods are robust

Reviewer #3: Fongwen et al, describe a strategy by which WHO's diagnostic technical advisory group for neglected tropical diseases (DTAG) assembled and charged a working group to develop target product profiles (TPPs) for new diagnostic tools to aid with the eradication of Yaws. The manuscript summarizes the work done by the working group and how it resulted in two separate TPPs, which are described in the paper.

**Results**

-Does the analysis presented match the analysis plan?

-Are the results clearly and completely presented?

-Are the figures (Tables, Images) of sufficient quality for clarity?

Reviewer #1: Minor revision. Figure 1 quality needs to be improved.

Reviewer #2: the results are clear

Reviewer #3: The authors provide a summary of the working group's recommendations in two Tables. The first table includes recommendations for the development of new diagnostic tools for diagnosis Yaws and table two provides guidance for how to develop testing strategies for azithromycin resistance (Table 2). There is no statistical analysis, since the paper is purely a description of the process by which the working group came up with their recommendations for testing for Yaws and for defining azithromycin resistance. The tables are clear in their present format, results are purely descriptive and aligned with the purpose of this manuscript.

**Conclusions**

-Are the conclusions supported by the data presented?

-Are the limitations of analysis clearly described?

-Do the authors discuss how these data can be helpful to advance our understanding of the topic under study?

-Is public health relevance addressed?

Reviewer #1: Minor revision, WHO reassured crteria should be moved to the dicussion.

Reviewer #2: the conclusions reflect an appropriate analysis of the data

Reviewer #3: Based on the existing literature and epidemiology of Yaws, the working group made certain assumptions about Yaws diagnostics and T. pallidum subs. pertenue azithromycin resistance in low and high risk populations and at different time periods in their respective journey towards eradication. The recommended TPPs have not yet been tested in any of the populations described. As a result this paper is not meant to be data driven and is purely hypothetical based on a number of well established assumptions.

**Editorial and Data Presentation Modifications?**

Reviewer #1: Accept

Reviewer #2: (No Response)

Reviewer #3: I recommend the authors add a statement in the discussion indicating that the recommendations will need to be validated in real life situations.

**Summary and General Comments**

Reviewer #1: This is a well-written manucsript describing target product profiles needed for yaws eradication efforts worldwide. I have minor comments for the authors to consider.

1. Neglected Tropical Diseases should be abbeviated in the Abstract

2. Line 6 of the abstract: The second "and" should be deleted 

3. The first and third sentences in the introduction seem contradictory with regard to endemicity of yaws in the tropics

4. Page 5: Since azithromycin resistance is a major concern, it's important to mention in which geographical location/s resistance has been reported. 

5. Page 5: Suggest changing "one of which" to "the latter" 

6. Page 6: Methods section, "The seven members" should read "The seven member" 

7. Figure 1: The text is illegible. 

8. Page 9: Suggest adding POC before lateral flow assays.

9. I suggest moving the statements on WHO reassured criteria to the discussion.

Reviewer #2: This revised submission is clear and concise.

Reviewer #3: Fongwen et al, provide a description of the process followed by the working group in coming up with two different target product profile development strategies which will ultimately result in control and eradication of Yaws. The manuscript is well written and the objectives are clear. Whether or not the proposed TPPs are feasible for high risk settings will require extensive field work. The bar which is set for diagnostics in Table 1 and Table 2 is very high and thus may not be achievable in real world situations.

PLOS authors have the option to publish the peer review history of their article (what does this mean?). If published, this will include your full peer review and any attached files.

Reviewer #1: No

Reviewer #2: Yes: Dr L Claire Fuller

Reviewer #3: Yes: Juan C Salazar

Figure Files:

Data Requirements:

Reproducibility:

References

---

## [Editor Report · Decision Letter 1]

9 Aug 2022

Dear Dr. Marks,

Thank you very much for submitting your manuscript "Diagnostics to support the eradication of yaws – Development of Two Target Product Profiles" for consideration at PLOS Neglected Tropical Diseases. As with all papers reviewed by the journal, your manuscript was reviewed by members of the editorial board and by several independent reviewers. The reviewers appreciated the attention to an important topic. Based on the reviews, we are likely to accept this manuscript for publication, providing that you modify the manuscript according to the review recommendations. 

The authors have addressed reviewer comments but the manuscript has some textual issues that should be addressed. As one example "latter????" on page 5. I also noted other typos (e.g., Ref 6) on p5. Please proof read carefully. 

""

Sincerely,

Justin D Radolf

Guest Editor

Richard Phillips

Section Editor

The authors have addressed reviewer comments but the manuscript has some textual issues that should be addressed. As one example "latter????" on page 5. I also noted other typos (e.g., Ref 6) on p5. Please proof read carefully. 

""

Figure Files:

Data Requirements:

Reproducibility:

References

---

## [Editor Report · Decision Letter 2]

15 Aug 2022

Dear Dr. Marks,

We are pleased to inform you that your manuscript 'Diagnostics to support the eradication of yaws – Development of Two Target Product Profiles' has been provisionally accepted for publication in PLOS Neglected Tropical Diseases.

Best regards,

Justin D Radolf

Guest Editor

Richard Phillips

Section Editor

---

## [Editor Report · Acceptance letter]

26 Aug 2022

Dear Dr. Marks,

We are delighted to inform you that your manuscript, "Diagnostics to support the eradication of yaws – Development of Two Target Product Profiles," has been formally accepted for publication in PLOS Neglected Tropical Diseases.

Best regards,

Shaden Kamhawi

co-Editor-in-Chief

Paul Brindley

co-Editor-in-Chief
